# Low frequency weak electric fields can induce structural changes in water

**Iman Rad**[ID]¤*, **Rainer Stahlberg, Kurt Kung, Gerald H. Pollack**

Department of Bioengineering, University of Washington, Seattle, Washington, United States of America

¤ Current address: Stem Cell Technology Research Center, Tehran, Iran
* radi@u.washington.edu, rad@strc.ac.ir

**Data Availability Statement:** All relevant data are within the manuscript and its Supporting Information files.

**Funding:** We were financially supported through a grant from the software AG foundation (SAGST)

## Abstract

Low frequency electric fields were exposed to various water samples using platinum electrodes mounted near the water surface. Responses were monitored using a spectro-radiometer and a contact-angle goniometer. Treatment of DI (deionized), EZ (Exclusion Zone), and bulk water with certain electromagnetic frequencies resulted in a drop of radiance persisting for at least half an hour. Compared to DI water, however, samples of EZ and bulk water showed lesser radiance drop. Contact-angle goniometric results confirmed that when treated with alternating electric fields (E = 600 ± 150 V/m, $f$ = 7.8 and 1000 Hz), droplets of EZ and bulk water acquired different charges. The applied electric field interacted with EZ water only when electrodes were installed above the chamber, but not beneath. Further, when DI water interacted with an electric field applied from above (E = 600 ± 150 V/m, $f$ = 75 Hz), its radiance profile became similar to that of EZ water. Putting these last two findings together, one can say that application of an electric field on DI water from above (E = 600 ± 150 V/m, $f$ = 7.8 to 75 Hz) may induce a molecular ordering in DI water similar to that of EZ water.

## 1. Introduction

In a previous report [1], we used a spectro-radiometer and an infrared camera to monitor the impact of applied electric currents in water. That work showed that application of extremely low frequency alternating currents induced structural changes in water, which were also confirmed by IR radiance spectra [1, 2]. Structural changes may also be reflected in changes of temperature: proximity of water to charged surfaces can trigger temperature drop, or even freezing [3, 4].

One of the challenges of the previous study was our inability to rule out the intervening impact of electrolysis during the application of electric current, i.e., when electrodes were immersed in water [5]. Therefore, in the current study, we attempted to determine whether external electric fields, i.e., with electrodes outside the water, and with relatively lower amplitudes, might cause similar structural changes.

Field strengths of $10^7$–$10^8$ V/m exist in nature, e.g. across the membrane of a resting cell with ~ 1–5 nm thickness [6, 7]. Theoretical predictions suggest that field strengths of $10^7$–$10^9$ V/m considerably diminish translational entropy of confined water [3, 4], which results in a

Foundation. The funders had no role in study design, data collection and analysis, decision to publish, or preparation of the manuscript.

**Competing interests:** The authors have declared that no competing interests exist.

temperature drop, or even freezing. An electric field with maximal amplitude of only ~ 750 V/m, as used in the current study, is not expected to make non-confined water freeze, but may diminish its temperature to some extent.

Here, we applied external electric fields on non-confined water samples to determine if any of those fields can cause a drop of temperature, or radiance. We also discuss potential mechanisms that may explain the observed results.

## 2. Material and methods

### 2.1. Experimental design

Three types of water were used in this study: DI (deionized), EZ (exclusion-zone) and bulk water. EZ and bulk waters, acquired from $4^{th}$-Phase Inc., were prepared by passing DI water through a Nafion tube (Perma Pure LLC), and extracting each of the two phases. Details of separation appear in the first figure of US7819259B2 patent [8].

Essentially, a Nafion tube was mounted between electrodes. EZ and bulk waters were formed in place, around the 6-mm long Nafion tube. After 10 minutes, an annular EZ layer commonly grew to a width of approximately 500 μm [9]. By examining the behavior of each of the above-mentioned waters directly in the Nafion setup, features of the EZ, as well as the bulk water beyond, could be directly assessed.

Since EZ formation next to Nafion is temperature dependent [10], after EZ and bulk waters were collected, we used a thermal stage to set and maintain the temperature. Experiments were run at 293, 283 and 273 K. Thermal control was set via a temperature control module (Omega-ette PV), as previously explained in detail [11].

### 2.2. Measurement setup

All experiments were conducted with purified water inside a shallow rectangular polypropylene dish (4.5 cm wide, 2.5 cm long and 0.5 cm high). The DI water was HPLC grade, 18.2 MΩ. cm$^{-1}$, obtained from a standard water-purification system (Diamond TII, Barnstead). The chamber was filled with three milliliters of either DI, EZ, or bulk water for different experiments. Before the experiment began, the chamber was filled, and left for 30 minutes for the water to become thermally stable. Then, exposure to the external electric field began. Water was exposed to alternating electric fields generated by a function generator (SRS, Model DS335) at a series of frequencies. The applied voltage was 6 ± 1.5 V, which created electric fields in the range of 450–750 V/m. This range is defined mathematically as a vector field associated with the distance between electrodes in a vacuum, and does not reflect the time that power is delivered. Exposure time extended over 30 minutes. This particular range of electric field strength was chosen since application of electric fields with amplitude of less than 750 V/m does not apparently freeze bulk water [4]. The field was created using two electrodes, installed either beneath or above the water-containing chamber (Fig 1A).

Platinum wires (Sigma-Aldrich, CAS: 7440-06-4, 5-cm length, 2-mm diameter) were used as electrodes. They were installed at a distance of 1 cm from one another, in the horizontal plane. The electrode pairs were set either 1 mm above the container-water surface or 1 mm beneath (Fig 1B and 1C, below the bottom of the water container). Schematic distribution of electric field on rectangular petri-dish is demonstrated using oPhysics: Interactive Physics Simulations (Fig 1D) [12].

The electrical energy needed for complete electrolysis of pure water in this apparatus, total conversion of water to $H_2$ and $O_2$, can be calculated to lie between 1.3–13 Watt-hours. We estimated the possible electrolysis in this study to be almost $10^9$ times lower than that, i.e., between $0.8–4.38 \times 10^{-9}$ Watt-hours.

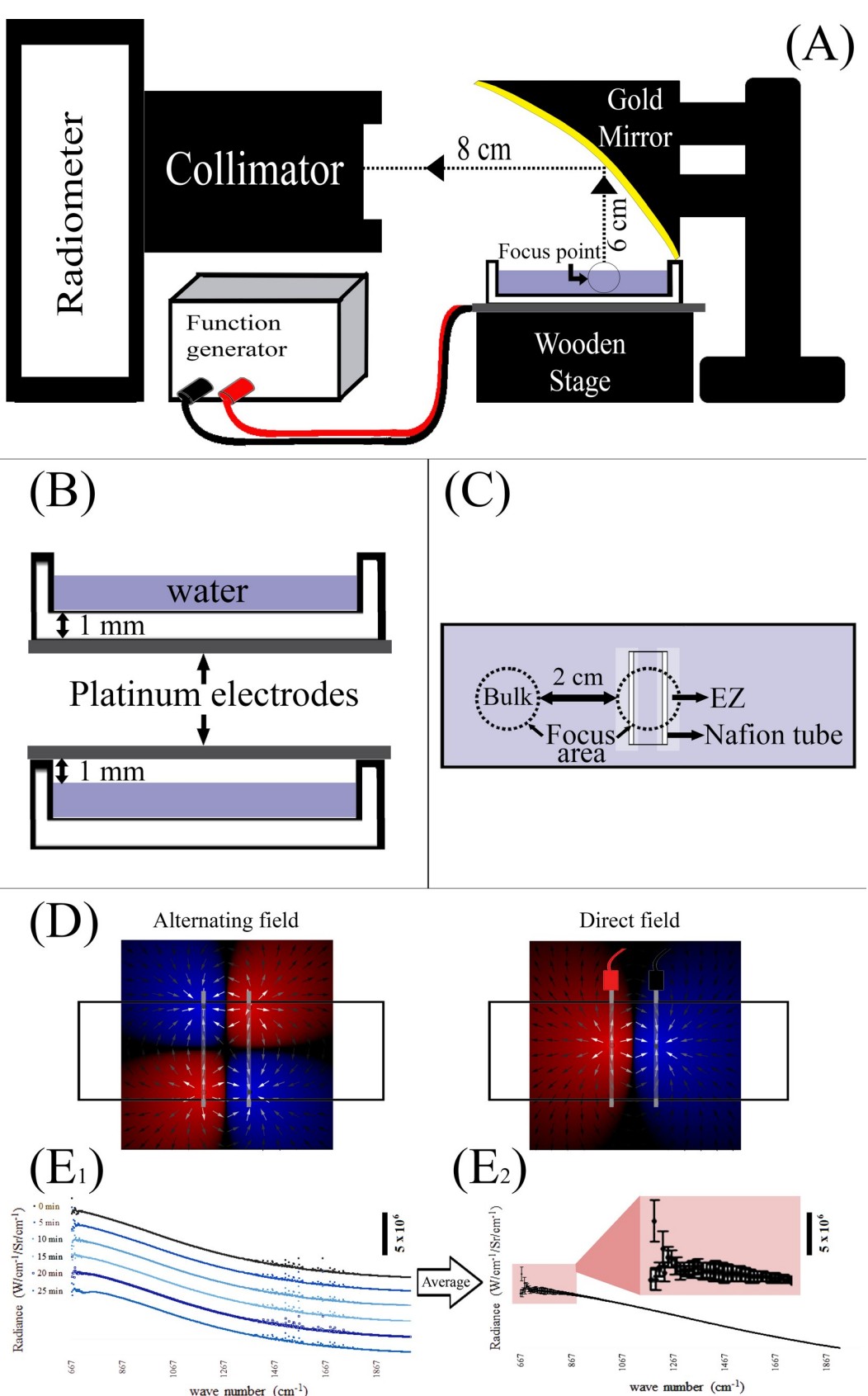

**Fig 1. Schematic views of measurement setup.** (A) Lateral view of spectro-radiometer and water chamber, the latter resting on a non-conductive wooden stage. Arrowheads denote the direction of travel of light from chamber to collimator. Red and black wires represent positive and negative poles, respectively. The schematic representation of the golden mirror's focus area, from lateral view, is confined to a circle. (B) Lateral view of electrodes that were installed beneath (upper panel) or above (lower panel) the water chamber. (C) Special case, in which EZ and bulk waters around Nafion tube were studied radiometerically, while neither an excitation nor an electrode was involved. (D) Schematic electric field distribution between electrodes. The overlapping shadow of petri-dish with electric field distribution is demonstrated. Red and blue colors represent positive and negative electric charge, respectively. (E$_1$) Six series of post-excitation data. Each data series (e.g. 0 min) is the average of 100 scans. The average of six data series is demonstrated in (E$_2$). The averaged points with standard deviation is "zoomed-in" in E2. None of the drawings is to scale.

## 2.3. IR-radiance recording setup

An MR170 spectro-radiometer (ABB Measurement & Analytics) was used to record the emission spectra in the wavelength range of 2–15 μm (5000–667 cm$^{-1}$) by the device's Fourier transforming software (FTSW500 Version 1.06).

The experimental setup and its software settings are explained in detail in a previous report [1]. Reproducibility of the measurement is correlated to the recorded noise: The higher the noise amplitude, the lower the expected reproducibility. In our experiments, calibration residuals were 10 to 20 times lower than the radiance amplitude, and could therefore be ignored. Radiance is expressed, in standard SI units, as watts per steradian per square centimeter per centimeter (W·Sr$^{-1}$·cm$^{-2}$·cm$^{-1}$).

Raw spectral data comes from averaging 100 measurements. The spectra are then calibrated against known sources of black-body radiation. Two black body references are used. Black bodies must have high emissivity and constant temperature. Hence, the palm of the hand and liquid nitrogen were set as references, representing the minimum and maximum thermal limits that were fed to the device's Fourier transforming software (FTSW500 Version 1.06)".

The schematic design of the experiment is shown in Fig 1. To record reproducible radiance values from spots of 1–2 mm diameter, we focused the IR emission using a concave gold mirror. The mirror's focus area is a circle of 2 mm diameter. The mirror reflected the emitted infrared radiation from the water surface into the spectro-radiometer's collimator tube.

**2.3.1 IR-radiance of EZ and bulk water in the presence of Nafion.** Besides use of the apparatus described above, we also wished to determine whether the interfacial water situated near the surface of a Nafion tube could be radiometrically differentiated from the bulk water, the latter situated at some distance away. By focusing the radiometer's gold mirror on the desired area, we could make this distinction. To this aim, we focused on the area between two walls of the Nafion tube that encompasses two EZs on the edges and bulk in between (i.e. the entire inside of the tube—Fig 1C). To measure bulk-water radiance, we placed the focus of the mirror 2 cm away from the Nafion wall. The mirror was similarly positioned for measuring the radiance of control water, but without the Nafion tube.

**2.3.2 IR-radiance of EZ, bulk, and DI water after treatment with electric field.** The properties of EZ and bulk water around the Nafion tube have been explored in previous studies [13, 14]. Here, we measured the impact of the electric-field on the EZ or bulk water. As first step, we compared the radiances of the field-treated EZ and bulk waters, with the corresponding untreated radiances as control. Since the electrodes were outside the water, they remained remote from the area of measurement.

## 2.4. Contact-angle measurement

The spreading of a water droplet on a surface is routinely measured by the "contact angle" method. Contact-angle (goniometric) measurements are useful in the evaluation of surface

macroscopic properties, such as surface energy and wettability [15]. Aqueous solutions in contact with surfaces experience strong Coulomb forces at their interfaces due to the charge interaction between the solution and surface [16–18]. Once the surface charge is known, the interfacial energy of such liquids facing solid surfaces can be estimated according to the measured static contact angle (θ) [19]. Therefore, knowing the charge of the surface in contact with the solution droplet, one can estimate the overall charge distribution of the droplet.

In order to monitor the spreading of a water droplet on positively and negatively charged surfaces, Teflon, Plexiglas, and glass surfaces were used. The aforementioned surfaces have static electric potentials of 7000 ± 500, -8000 ± 500 and 15 ± 10 V/m, respectively, which marks them as positively, negatively and almost neutrally charged surfaces that result in corresponding contact angles of 95.3, 66.7 and 40 degrees, respectively, in a spreading DI water droplet [15]. Static electric potential of the surfaces was measured by Static Locater/Meter Model 9000 (PN 90930–10001) non-contacting voltmeter (S1 Data).

To measure contact angle (θ), a CCD camera was set beside a horizontally positioned sheet with the droplet in the middle. The captured side view of each water droplet was recorded, and then analyzed. The contact angle "θ" of each water droplet was calculated using Image J software. Contact angle measurements were repeated five times and averaged.

## 2.5 Data analysis

Raw spectral data emerge as the average of 100 scans, which took 10–20 seconds to acquire, as described previously [1]. Radiometric sampling was performed at five-minute intervals during 30 minutes of measurement, which gave six sets of data. Raw spectral data were calibrated according to the emissivity references at each time interval. Six sets of "calibrated data" at each time were then averaged and reported as either "control", "excitation" or "post-excitation" data series, which are represented in Figs 2 to 6. To avoid image clutter in Figs 2 to 6, the deviation from average in each point of the data series, which is represented by error bars, has been removed. Besides, the data acquisition process is explained graphically in Fig 1E. The averaged data series (Fig 1E$_2$) are then compared statistically. Both radiometric and contact angle measurements were statistically analyzed using one-way ANOVA, and cases with calculated p-value of less than 0.05 were considered as significant.

## 3. Results

We used a spectro-radiomenter to measure the microscopic structural variations of water, while macroscopic charge distribution was assessed via contact-angle measurements. It was hoped that the results of these combined methods could bridge the gap between microscopic and macroscopic behaviors of water droplets.

In order to first rule out the possibility that radiation from the electrodes themselves might interfere with the radiometric measurements, an electrical potential was applied between the electrodes, installed either beneath or above the empty chamber. Voltages were applied for 30 minutes, while radiance was measured both during that period and for 30 minutes after the voltage application had ceased. Radiometric measurements of the electrodes alone (no water sample was involved) showed that when voltage was applied, no significant electrode radiance was observed, which implies that voltage application by the electrodes should not interfere with the radiometric measurements.

We then treated the water with electric fields of various frequencies. Fields were turned on for 30-minute periods. Radiometric measurements were made during the 30-minute period and for another 30 minutes thereafter. The latter period is referred to as the "post-excitation" phase. The entire time course, including excitation and post-excitation, was limited to 60

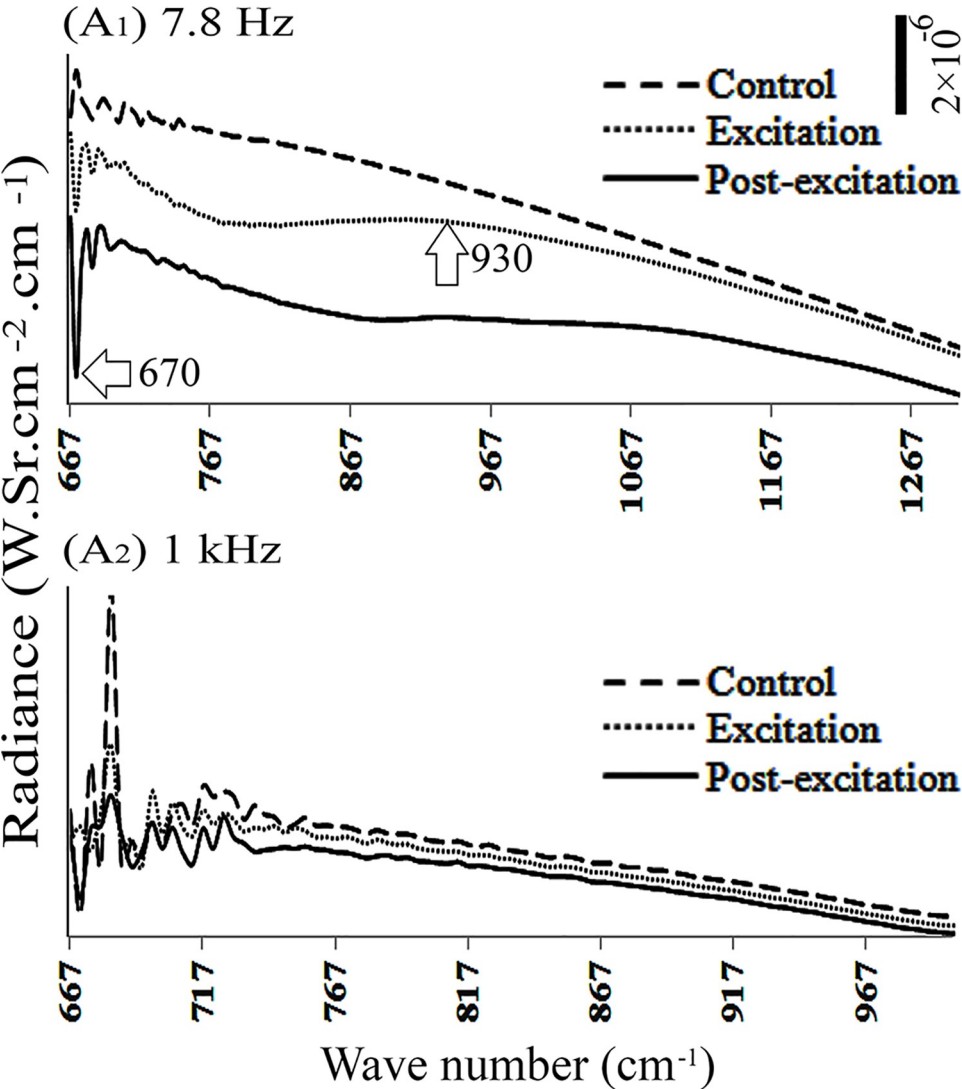

**Fig 2. Radiance spectra of DI water during and after application of low-frequency electric field, with electrodes installed beneath the chamber.** The amplitude scale is shown by a vertical bar at the right top of the figure, and the vertical axes have the same scale. Horizontal arrow denotes a region that has a significant variation of amplitude compared to the control.

minutes to avoid excessive evaporation, which would diminish chamber-water volume and hence potentially affect the results. Each radiometric experiment was repeated three to five times to ensure reproducibility.

### 3.1. Application of electric fields to DI water

These experiments focused on standard DI water. The DI water alone was subjected to 30 minutes of "excitation". The "control" water received no excitation.

When frequencies of 0, 4, 11, 16, 50, 75, 100 and $10^5$ Hz were applied from beneath, the radiance of DI water remained unchanged during control, excitation, and post-excitation periods. However, similar application of 7.8 and $10^3$ Hz (1 kHz) caused a reduction of water radiance relative to the control, at specific wave numbers. They ranged from 667–1300 and 667–

1000 cm$^{-1}$, either during the excitation or post-excitation periods, respectively (Fig 2A$_1$ and 2A$_2$).

In another set of experiments, electric fields with frequency of 0, 4, 7.8, 11, 16, 50 and 10$^5$ Hz were applied to DI water from above. Results showed no change in radiance during excitation and post-excitation periods. However, application of 75, 10$^2$ and 10$^3$ Hz frequencies from above dropped the water radiance, both during excitation and post-excitation periods (Fig 3A$_1$ to 3A$_3$).

## 3.2. Application of electric field on bulk water

To obtain "bulk" water, we used Nafion tubes. The water that was collected from just inside the Nafion-tube wall (within 100 μm proximity) was taken as EZ water, while the water passing through the core of the tube (more than 500 μm away from the wall) was considered bulk water.

Frequencies were chosen based on earlier observations. Since application of electric fields to DI water showed a substantial radiance drop at frequencies of 7.8, 75, 10$^2$ and 10$^3$ Hz, these four frequencies were chosen for testing on preparations of "EZ" and "bulk" water.

Application of electric field with frequency of 7.8 Hz. on bulk water either from above or beneath had no impact on the water radiance. When the applied frequency was increased to 75 Hz, the excitation and post-excitation radiance dropped in the spectral region 667–1000 cm$^{-1}$. A similar drop was observed when a frequency of 10$^3$ Hz applied from above (Fig 4).

## 3.3. Application of electric field to EZ water

Samples of "EZ water" were prepared the day prior to the experiment. The "EZ" samples are never completely pure EZ, and may be partly mixed with some bulk water.

Electric fields with frequencies of 7.8, 75, 10$^2$ and 10$^3$ Hz were applied to EZ water from beneath and above, independently. Only when the frequency of 7.8 Hz was applied from above did the radiance drop during excitation and post-excitation periods (Fig 5). Other treatments produced no observable change.

## 3.4. Effect of temperature on radiance

Positioning of a Nafion tube in DI water leads to the formation of EZ and bulk water (Fig 1C). With the tube in place, we measured radiance of EZ water around the Nafion tube, and also the bulk water—at different temperatures. By this method, the impact of temperature on the radiance of EZ and bulk water could be differentiated more decisively. For example, at 293 K, in the spectral region of 667–1000 cm$^{-1}$ the radiance of both EZ and bulk water were significantly lower than that of DI water (Fig 6A and 6B). In other spectral regions, the radiance of the two did not differ.

When the temperature diminished to 283˚K, the radiance of EZ water at 667–2000 cm$^{-1}$ was lower than that of bulk and DI waters (Fig 6C$_1$–6C$_3$). Moreover, with further water-temperature decrease to 273 K, the radiance of control, EZ and bulk waters became non-differentiable. Therefore, EZ and bulk water are more profoundly differentiable at higher temperature than a lower temperature.

## 3.5. Variation of charge in different treated water droplets

Here we used 50 μL droplets of EZ, bulk, and DI water. After being treated with 7.8 Hz and 1000 Hz electric fields, EZ and bulk water droplets of the same volume were tested by contact-angle measurement. Measurement of each type of water droplet was repeated five times.

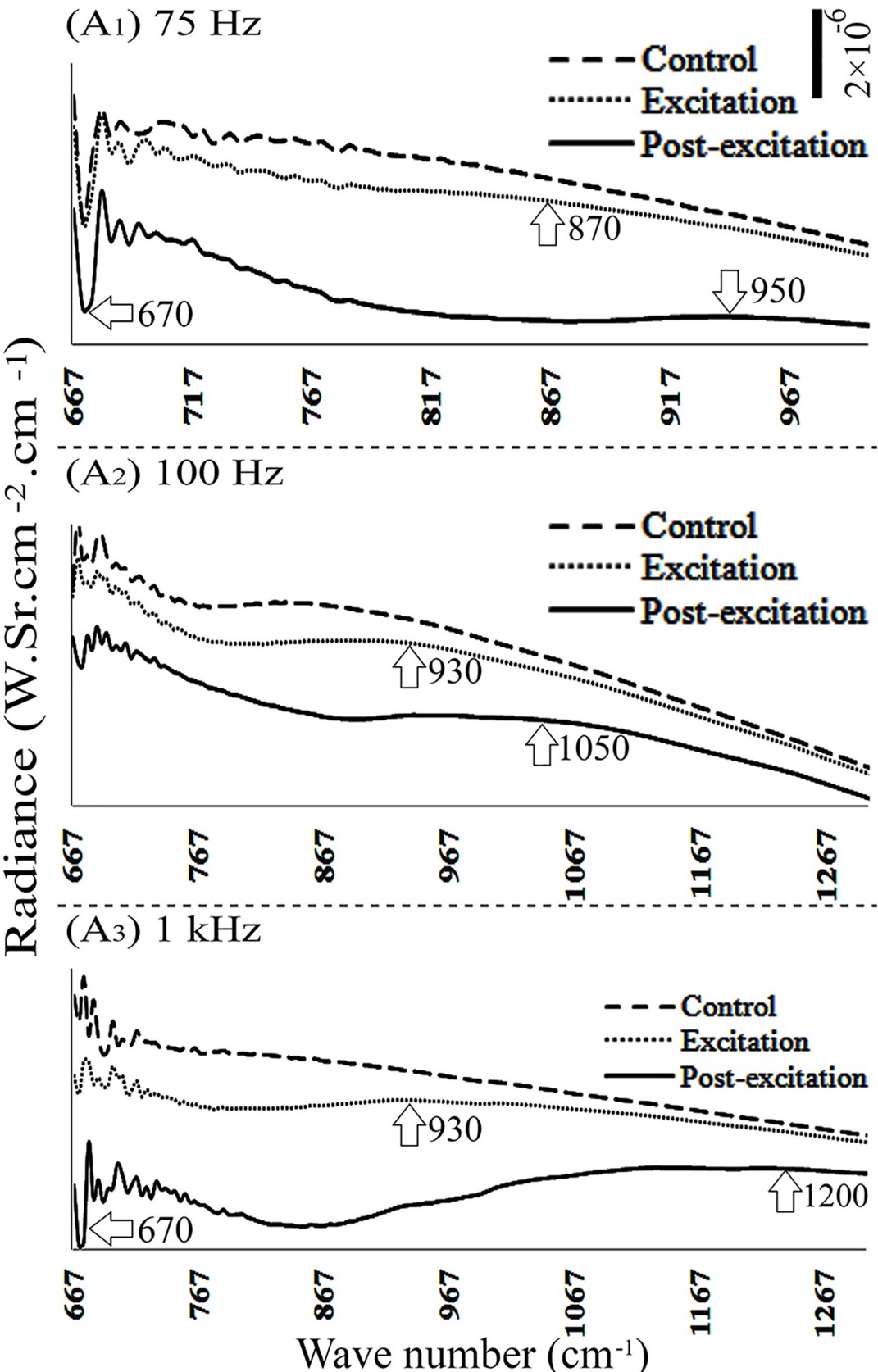

**Fig 3. Radiance spectra of DI water,** obtained when electric field was applied from above the chamber, using different frequencies (A₁–A₃).

Droplets of untreated water, EZ and bulk, were considered as "EZ control" and "bulk control", respectively. Experimental samples were treated, respectively, with electric fields of 7.8 Hz ("EZ 7.8 Hz") and 1000 Hz ("bulk 1000 Hz"), (Fig 7A). As mentioned in section 3.2., frequencies were chosen based on earlier observations.

Measurement of the contact angle "θ", acquired from different types of water droplets on the neutrally charged surface of the glass, showed no notable differences among "EZ control", "EZ 7.8 Hz", "Bulk control" and "Bulk 1000 Hz" (Fig 7B).

On the positively and negatively charged surfaces, such as Teflon and Plexiglas, comparison of sitting water droplets showed that "Bulk control" droplet made a larger contact angle "θ" compared to the "Bulk 1000" droplet (Fig 7C).

## (A) EZ an bulk water around the Nafion tube

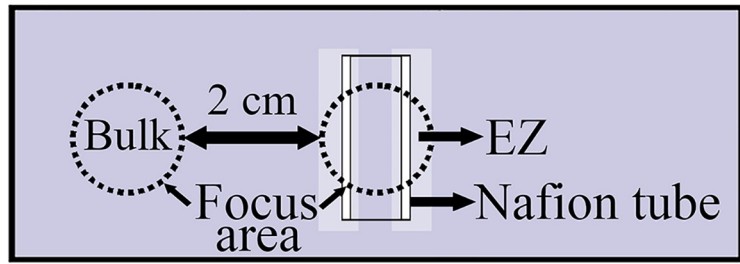

## (B) 75 Hz, electric field from beneath

## (C) 1 kHz, electric field from above

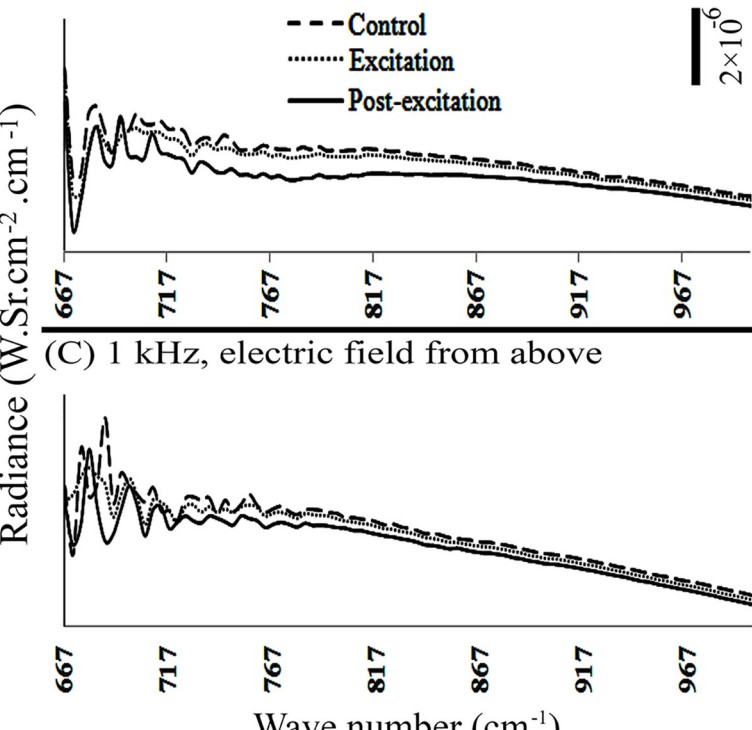

**Fig 4. Radiance of "bulk water" during and after exposure to electric field.** The measurements were made with electrodes (A) placed either beneath (B) or above (C) the chamber.

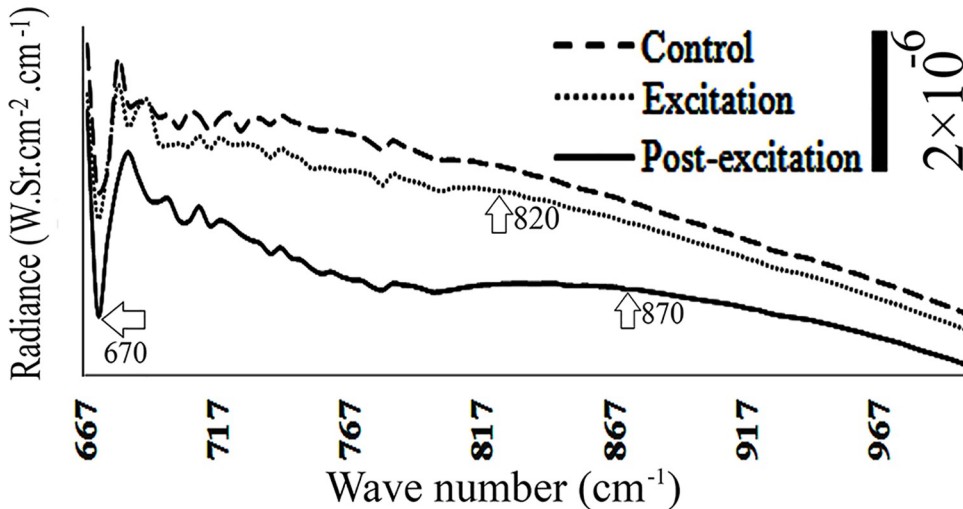

**Fig 5. EZ water radiance profile during and after exposure to 7.8 Hz electric field.** Electrodes placed above the chamber.

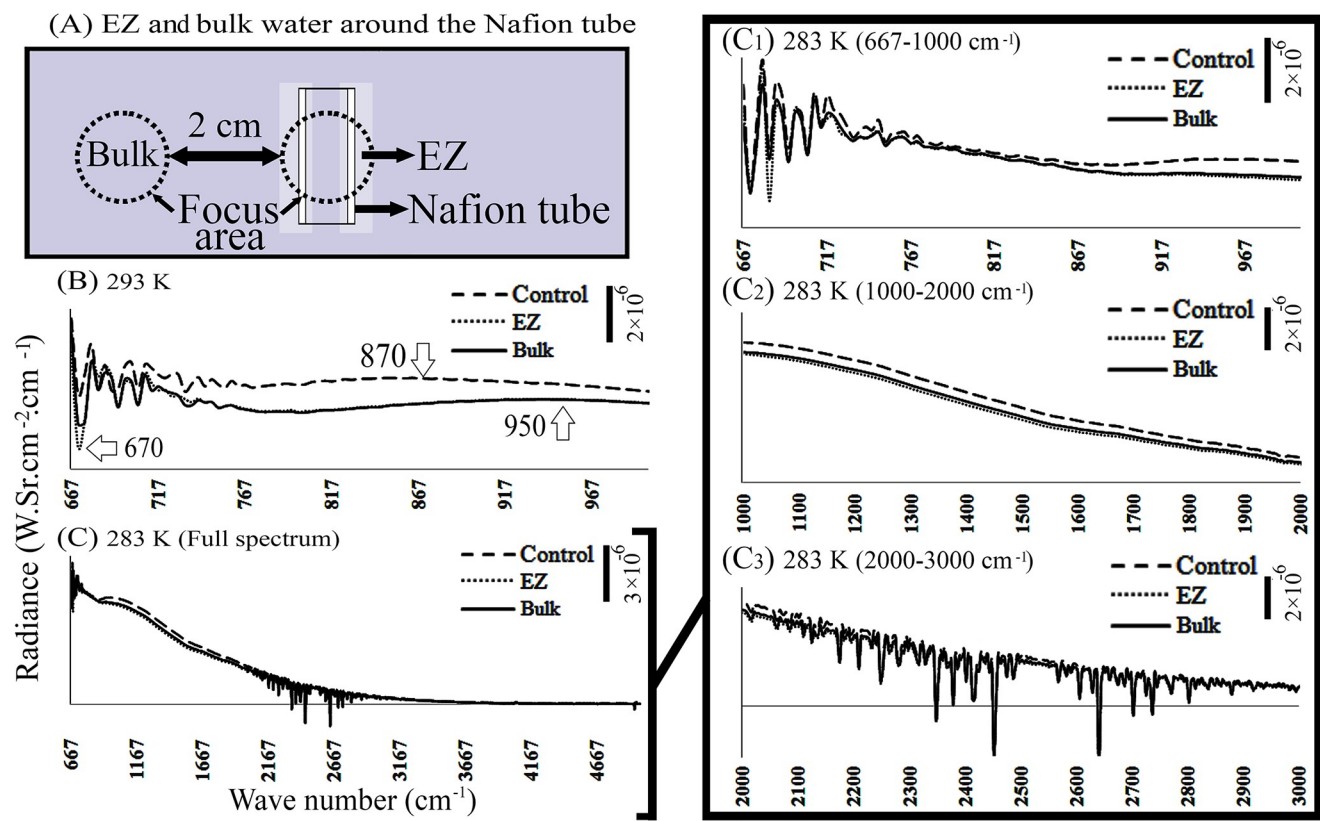

**Fig 6. Radiance of water during and after exposure to Nafion tube.** (A) Schematic top view of water chamber with a 6-millimeter long Nafion tube placed at the center of the water chamber. The pale area around the tube represents EZ water. "Control" represents the measurement taken before the Nafion tube had been placed in the water. (B) Radiance of EZ and bulk water at room temperature (293 K). (C) Radiance of EZ and bulk water at 283 K is shown with higher magnification at different wave-number regions (C$_1$ to C$_3$).

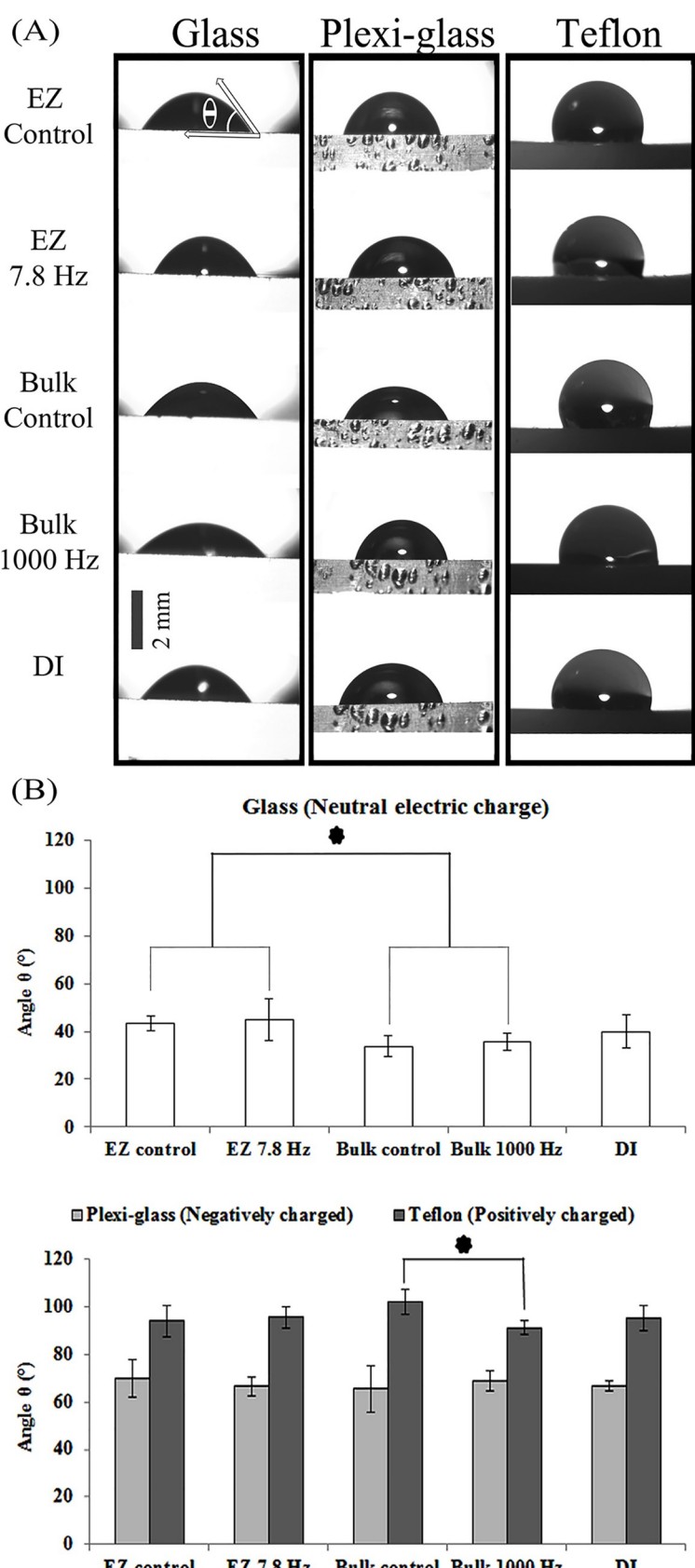

**Fig 7. Contact-angle measurement of five droplet types.** (A) Comparision of droplet shapes and curvatures on uncharged, positively charged, and negatively charged surfaces. (B) Comparison of measured contact angle of water droplets. The stars above the graphs in section B, represent the significant differences among cases or groups.

## 4. Discussion

Application of electric fields of certain frequencies diminished water radiance. The fields, of magnitude 600 ± 150 V/m, near the electrodes were applied either from above or beneath the water-containing chamber. Deionized water samples were treated with frequencies of 0, 4, 7.8, 11, 16, 50, 75, $10^2$, $10^3$ and $10^5$ Hz. (Note that the frequency of 7.8 Hz, the Schumann frequency, was used instead of 8 Hz, since this frequency is reported to have a role in transduction of biological information [2, 20, 21]). During control, excitation, and post-excitation periods, we assessed potential structural changes by measuring the water's IR radiance. Besides, EZ and bulk water droplets that were pre-treated with electric fields also showed different radiance and charge distribution. In the following section, we will discuss electric field impacts in detail.

### 4.1. Radiometric findings

Depending on the position of the electrodes, beneath vs. above, and the frequency of the applied electric field, water samples' radiance changed differently. For instance, when the electric field (E = 600 ± 150 V/m) was applied to DI water from above with frequencies of 75, $10^2$ or $10^3$ Hz, or to bulk water with frequency of $10^3$ Hz, water radiance diminished within the 677–1000 cm$^{-1}$ range. Similar impact was observed upon application of the electric field from beneath, with frequencies of 7.8 and $10^3$ Hz on DI water, or with frequencies of 75 and 7.8 Hz on bulk and EZ waters, respectively (Figs 2, 3 and 5). Other frequencies produced no significant changes in radiance.

Thus, application of electric fields from beneath imposed a radiance-diminishing impact on all water-sample types. However, application of electric field from above modified only DI and bulk-water radiance, and not EZ radiance.

When the field was applied, both DI and bulk waters showed diminished radiance independent of field direction, while EZ water showed diminished radiance only when the field was applied from above (Fig 5). Therefore, it appears that EZ water may be more reluctant to acquire further structural modification after exposure to the electric field. A possible interpretation is that EZ water is more stable than DI or bulk water. Since it is already ordered, any additional ordering caused by electric-field application may be difficult to achieve.

When liquid water is subject to an inhomogeneous electric field ($\nabla^2 E_a \approx 10^{10}$ V/m$^2$), enhancement of the vibrational collective modes, hindered rotational freedom of long range dipole–dipole interactions, and retardation of heat flow occurs, which eventually generates a chemical potential gradient in electrified water [22]. In other words, liquid water stressed by an electric field, exhibits primary signs of phase transition throughout the entire volume of liquid [23]. Previously, it was reported that water molecules tend to get ordered because of diminution in their translational entropy [1, 24, 25]. Diminished translational entropy (i.e. librational mode) is also detectable as a sharpened peak in radiometric measurements around 600 cm$^{-1}$, and a temperature drop from 300˚K to 240˚K [26]. In this study, application of an electric field resulted in diminished translational entropy and temperature drop in DI and EZ water (Figs 2, 3, 5 and 6), although it was not significant in bulk water (Fig 4B and 4C).

### 4.2. Insights from contact-angle measurements

Contact-angle measurement comes with the expectation that positively charged water droplets have the highest contact angle on a positively charged surface, and the smallest contact angle

on a negatively charged surface, and vice-versa for negatively charged droplets. Thus, depending on the charge of surface and water sample, a droplet acquires a specific contact angle "θ" [27]. If both the water droplet and the surface have same electrical charge, θ will change from 90 degrees toward higher angles due to repulsion. Otherwise, the θ angle will drop from 90 degree toward lower angles.

With this understanding, it appears that both "bulk control" and "bulk 1000 Hz" droplets had less positive charge content than both the "EZ control" and "EZ 7.8 Hz" droplets (Fig 7B). The "bulk control" and "bulk 1000 Hz" droplets did not show any difference when they sat on a glass surface (Fig 7B). However, when both "bulk control" and "bulk 1000 Hz" sat on a Teflon surface, it appeared that the "bulk control" droplet was more positively charged than "bulk 1000 Hz" droplet (Fig 7C).

Putting all the contact-angle results together, we can conclude that bulk and EZ water droplets have different charge content, making them positively or negatively charged with respect to one another. Treatment of EZ water with an electric field ($f$ = 7.8 Hz) makes the droplet more positive, while application of electric field on bulk water ($f$ = 1000 Hz) makes it more negative. Hence, these fields appear to confer differing charges to water.

## 5. Conclusion

Treatment of EZ and bulk-water droplets with alternating electric fields altered their charge. Mainly, they appeared to gain positive charge, in the following order: bulk-control < bulk-1000 Hz < EZ-control, DI and EZ-7.8 Hz. Putting the radiometric and contact-angle goniometric results together led us to conclude that EZ water has higher structural stability and greater tendency to absorb positive charge than DI and bulk water.

On the other hand, our principal results show that application of an electric field from above, with frequency of 75 Hz on DI water, changes the water's molecular structure in a way that its radiance profile shows similarity toward radiance profile of EZ water, while other frequencies had no such impact. In other words, that frequency would result in "relatively ordered water", which is also a signature feature of EZ water. However, how the molecular ordering of EZ and "75 Hz exposed water" are similar or coincidental is a matter for future study.

## Supporting information

**S1 Data.**
(XLSX)

**S1 Graphical abstract.**
(TIF)

## Author Contributions

**Conceptualization:** Iman Rad, Gerald H. Pollack.

**Data curation:** Gerald H. Pollack.

**Formal analysis:** Iman Rad.

**Investigation:** Iman Rad, Rainer Stahlberg.

**Methodology:** Kurt Kung.

**Resources:** Rainer Stahlberg.

**Supervision:** Gerald H. Pollack.

**Writing – original draft:** Iman Rad.

**Writing – review & editing:** Iman Rad, Gerald H. Pollack.

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
