## [Decision Letter · Decision Letter 0]

4 Sep 2021

PONE-D-21-22146

Low frequency weak electric fields can induce structural changes in water

PLOS ONE

Dear Dr. Rad,

Thank you for submitting your manuscript to PLOS ONE. After careful consideration, we feel that it has merit but does not fully meet PLOS ONE’s publication criteria as it currently stands. Therefore, we invite you to submit a revised version of the manuscript that addresses the points raised during the review process.

We look forward to receiving your revised manuscript.

Kind regards,

Wei-Chun Chin

Academic Editor

PLOS ONE

2. In your Methods section, please include additional information about the methods used in the study, instead of referring to previous publications, in order to ensure that the study is fully reproducible by another researcher.

“We thank the Software AG foundation (SAGST) for the support of this work.”

Additional Editor Comments (if provided):

please address comments/suggestions from reviewers and revise your manuscript.

Reviewers' comments:

Reviewer's Responses to Questions

**Comments to the Author**

1. Is the manuscript technically sound, and do the data support the conclusions?

Reviewer #1: Yes

Reviewer #2: Yes

Reviewer #3: Partly

2. Has the statistical analysis been performed appropriately and rigorously? 

Reviewer #1: I Don't Know

Reviewer #2: No

Reviewer #3: Yes

3. Have the authors made all data underlying the findings in their manuscript fully available?

Reviewer #1: Yes

Reviewer #2: Yes

Reviewer #3: Yes

4. Is the manuscript presented in an intelligible fashion and written in standard English?

Reviewer #1: Yes

Reviewer #2: Yes

Reviewer #3: Yes

5. Review Comments to the Author

Reviewer #1: The paper provide new insights into the interaction of electromagnetic fields with aqueous systems. It also provide grounded experimental evidence of the possible existence of different frequency windows of interaction.The latter could account for possible therapeutic effect of specific delivered electromagnetic signals trough aqueous systems embedded into cells, organs and tissues. According to the latest purpose I suggest, in further future, experiment to perform the experiments in the physiological temperature range of humans and other living organisms including infectious agents.

Reviewer #2: The manuscript lacks sufficient statistical treatment. In various figures (graphs), it would be necessary to add a statement of what the presented curve represents: a statistical average of measurements or the best results.

There are also minor issues that should be clarified. They are presented in the revised pdf version of the manuscript.

Reviewer #3: Water molecules play essential roles in most of biological process, and the findings of structured water are really interesting and may contribute to our current understanding in biology. However, due to the diverse states of water molecule, the complex molecular interactions remain largely unknown. In this study, Rad et.al showed the low frequency weak electric fields can induce structural changes in water. These findings are very interesting, and may further be applied for various biological manipulation through the alteration of water structure.

This manuscript contains a set of interesting information that should attract the attention of a broad readership. I recommend publication with revisions (see below), which are mainly concerned with clarifications.

1. “Putting these last two findings together, one can say that application of an electric field on DI water from above (E = 600 ± 150 V/m, f = 7.8 to 75 Hz) may induce a molecular ordering in DI water similar to that of EZ water”

One of the unique features of EZ water is the capacity to repelling solutes. It seems not convincing to claim the molecular ordering in DI water similar to that of EZ water, merely based on radiance profile.

2. “All experiments were conducted with purified water inside a shallow rectangular polypropylene dish (4.5 cm wide, 2.5 cm long and 0.5 cm high).”

Please clarify the reason for using purified water in the experiment, since most biochemical reactions are performed in unpurified condition.

Does the shape of dish influence the results here? Does the height of DI water change the pattern of radiance readout? Also, the influences of shape of electrode and the spatial distribution of electric field on specimens may be interesting to readers. Please clarify in the discussion section.

3. “Raw spectral data emerge as the average of 100 scans, which took 10-20 seconds to acquire”

Please clarify the average of 100 scans come from single experiment or three-independent experiments.

4. “Each radiometric measurement was repeated three to five times…”

There is no data shows the replications and the variation between each experiment. In addition, please revise figure caption with more experiment details.

5. “…the excitation and post-excitation radiance dropped… “

Is the radiance drop significantly different from the control? Without statistical interpretation, it is difficult to determine the difference.

Moreover, compared the data in figure -4b and -4c, the spectrum of control group are different; especially, there is an distinguishing peak around 690 (cm-1) in 4c. Moreover, previous studies showed the repelling capacity of EZ water decreases with time. Did the electric field induced alteration of water last for more than 60 min?

6. PLOS authors have the option to publish the peer review history of their article (what does this mean?). If published, this will include your full peer review and any attached files.

Reviewer #1: No

Reviewer #2: **Yes: **Prof. Dr. Igor Jerman

Reviewer #3: No

---

## [Author Response · Author response to Decision Letter 0]

28 Oct 2021

Respond to reviewer and editor comments is uploaded as a file named as "Response to reviewers-3".

---

## [Decision Letter · Decision Letter 1]

22 Nov 2021

Low frequency weak electric fields can induce structural changes in water

PONE-D-21-22146R1

Dear Dr. Rad,

We’re pleased to inform you that your manuscript has been judged scientifically suitable for publication and will be formally accepted for publication once it meets all outstanding technical requirements.

Kind regards,

Wei-Chun Chin

Academic Editor

PLOS ONE

Additional Editor Comments (optional):

Reviewers' comments:

Reviewer's Responses to Questions

**Comments to the Author**

1. If the authors have adequately addressed your comments raised in a previous round of review and you feel that this manuscript is now acceptable for publication, you may indicate that here to bypass the “Comments to the Author” section, enter your conflict of interest statement in the “Confidential to Editor” section, and submit your "Accept" recommendation.

Reviewer #1: All comments have been addressed

Reviewer #2: (No Response)

Reviewer #3: (No Response)

2. Is the manuscript technically sound, and do the data support the conclusions?

Reviewer #1: Yes

Reviewer #2: Yes

Reviewer #3: Yes

3. Has the statistical analysis been performed appropriately and rigorously? 

Reviewer #1: Yes

Reviewer #2: Yes

Reviewer #3: Yes

4. Have the authors made all data underlying the findings in their manuscript fully available?

Reviewer #1: Yes

Reviewer #2: Yes

Reviewer #3: Yes

5. Is the manuscript presented in an intelligible fashion and written in standard English?

Reviewer #1: Yes

Reviewer #2: Yes

Reviewer #3: Yes

6. Review Comments to the Author

Reviewer #1: (No Response)

Reviewer #2: The manuscript is an important contribution to disentangle the mystery of EZ water and the bulk water that is its "negative". Even if there may be some other interpretations of the results, your measurements clearly show that water can be changed in a significant manner and that these changes may last a considerable time. The research also shows that we are only et the beginning of understanding a complex interaction between electric field and water, which may be very important also for a new understanding of life.

Reviewer #3: (No Response)

7. PLOS authors have the option to publish the peer review history of their article (what does this mean?). If published, this will include your full peer review and any attached files.

Reviewer #1: No

Reviewer #2: **Yes: **Prof. Dr. Igor Jerman

Reviewer #3: No

---

## [Editor Report · Acceptance letter]

24 Nov 2021

PONE-D-21-22146R1 

Low frequency weak electric fields can induce structural changes in water 

Dear Dr. Rad:

I'm pleased to inform you that your manuscript has been deemed suitable for publication in PLOS ONE. Congratulations! Your manuscript is now with our production department. 

Kind regards, 

on behalf of

Dr. Wei-Chun Chin 

Academic Editor

PLOS ONE